# Recombinase Polymerase Amplification Combined with Lateral Flow Dipstick Assay for the Rapid and Sensitive Detection of *Pseudo-nitzschia multiseries*

**DOI:** 10.3390/ijms25021350

**Published:** 2024-01-22

**Authors:** Yuqing Yao, Ningjian Luo, Yujie Zong, Meng Jia, Yichen Rao, Hailong Huang, Haibo Jiang

**Affiliations:** 1School of Marine Sciences, Ningbo University, Ningbo 315211, China; 216003682@nbu.edu.cn (Y.Y.); 2111091025@nbu.edu.cn (N.L.); 216003820@nbu.edu.cn (Y.Z.); 216002513@nbu.edu.cn (M.J.); 216004771@nbu.edu.cn (Y.R.); 2Southern Marine Science and Engineering Guangdong Laboratory (Zhuhai), Zhuhai 519080, China

**Keywords:** *Pseudo-nitzschia multiseries*, recombinase polymerase amplification, lateral flow dipstick, detection, internal transcribed spacer

## Abstract

The harmful algal bloom (HAB) species *Pseudo-nitzschia multiseries* is widely distributed worldwide and is known to produce the neurotoxin domoic acid, which harms marine wildlife and humans. Early detection and preventative measures are more critical than late management. However, the major challenge related to early detection is the accurate and sensitive detection of microalgae present in low abundance. Therefore, developing a sensitive and specific method that can rapidly detect *P. multiseries* is critical for expediting the monitoring and prediction of HABs. In this study, a novel assay method, recombinase polymerase amplification combined with lateral flow dipstick (RPA-LFD), is first developed for the detection of *P. multiseries*. To obtain the best test results, several important factors that affected the amplification effect were optimized. The internal transcribed spacer sequence of the nuclear ribosomal DNA from *P. multiseries* was selected as the target region. The results showed that the optimal amplification temperature and time for the recombinase polymerase amplification (RPA) of *P. multiseries* were 37 °C and 15 min. The RPA products could be visualized directly using the lateral flow dipstick after only 3 min. The RPA-LFD assay sensitivity for detection of recombinant plasmid DNA (1.9 × 10^0^ pg/μL) was 100 times more sensitive than that of RPA, and the RPA-LFD assay sensitivity for detection of genomic DNA (2.0 × 10^2^ pg/μL) was 10 times more sensitive than that of RPA. Its feasibility in the detection of environmental samples was also verified. In conclusion, these results indicated that the RPA-LFD detection of *P. multiseries* that was established in this study has high efficiency, sensitivity, specificity, and practicability. Management measures made based on information gained from early detection methods may be able to prevent certain blooms. The use of a highly sensitive approach for early warning detection of *P. multiseries* is essential to alleviate the harmful impacts of HABs on the environment, aquaculture, and human health.

## 1. Introduction

Harmful algal blooms (HABs) are destructive biological events resulting from the rapid production and accumulation (retention, physical transport, and behavior) of microalgae and other organisms (such as *Mesodinium rubrum*) [1,2]. The number and scale of HABs are on the rise globally due to the intensification of water eutrophication and global climate change [3]. In addition, more than 70 HAB species produce toxins that pose threats to both wild and cultured marine animals and human consumers through the food chain [4,5]. Alarmingly, algal toxins account for 15% of annual poisoning-related deaths. Over 70 species of microalgae are known to produce toxins, with 28 of these species being found in China [6,7].

Among the HAB species within the genus *Pseudo-nitzschia* (Bacillariophyta), several are capable of producing the neurotoxin domoic acid (DA), with at least twelve species identified as toxin producers [8,9]. The neurotoxin DA has the potential to induce amnesic shellfish poisoning (ASP), with the first recorded ASP incident occurring in 1987 [10,11,12]. Subsequent investigations confirmed the origin of this toxic substance in the genus *Pseudo-nitzschia*, and since then, DA-producing *Pseudo-nitzschia* species have been discovered worldwide, including the HAB species of *Pseudo-nitzschia multiseries* [5,13,14,15]. The diatom genus *Pseudo-nitzschia* has been associated with ASP events globally and is one of the key harmful microalga groups in the Guangdong coastal waters, situated in the northern South China Sea (SCS) [9]. Recently, DA has frequently been detected in Chinese shellfish. The amount of shellfish produced in China for aquaculture in 2021 was 15.46 Mt. Shellfish are the dominating cultured organisms and a potential vector for algal toxins [8]. Given the alarming increase in HABs in China, increased attention should be paid to all the HABs that have the potential to cause DA contamination [16]. Hence, to provide a rapid emergency response to the HABs of *Pseudo-nitzschia* species, including *P. multiseries*, it is necessary to establish an accurate and rapid point-of-care detection approach [17].

Due to the expertise and time that are required for the microscopic discrimination of species, molecular methods that monitor the environmental concentrations of *Pseudo-nitzschia* provide a rapid alternative for early bloom detection and toxin accumulation prediction. The main molecular detection methods for HAB species have included polymerase chain reaction (PCR) [18], quantitative PCR (qPCR) [19,20,21], enzyme-linked immunosorbent assay [22], rolling circle amplification [23], helicase-dependent amplification [24], nucleic acid sequence-based amplification [25], loop-mediated isothermal amplification (LAMP) [26], recombinase-aided amplification (RAA) [27], and recombinase polymerase amplification (RPA) [28], among others [29]. Currently, the detection of *P. multiseries* is mainly divided into two categories. The first involves traditional morphological examination, which is a very common and straightforward detection method, but there is a high degree of subjectivity that necessitates professionals with extensive experience for identification [22]. The second is molecular systematics, such as qPCR, which has a straightforward procedure but requires specific equipment, making on-site field detection challenging [20].

Delaney et al. (2011) developed a quantitative reverse transcription PCR (qRT-PCR) assay for the detection of *P. multiseries* targeting the ribulose-1,5-biphosphate carboxylase/oxygenase small subunit (rbcS) gene [20]. The rbcS qRT-PCR assay is useful for the detection and enumeration of low concentrations of *P. multiseries* in the environment. Nevertheless, although molecular approaches have increased in their efficiency and speed of detection, they still have drawbacks, including specialized equipment requirements, the time that is required, and costliness. Additionally, isothermal nucleic acid amplification, without the use of complex temperature cycling systems, is well-suited for field rapid detection (point-of-care testing, POCT) and is becoming increasingly feasible due to ongoing advancements in molecular biology [29,30,31].

Recombinase polymerase amplification is a highly noteworthy and increasingly popular molecular detection method [32,33,34,35]. It is an isothermal nucleic acid amplification approach that uses three proteins and runs under isothermal conditions of about 37 °C [36]. Notably, it requires no specialized tools for sample preparation [37]. Moreover, it is not difficult to design the primers, and only two primers are needed to provide a high level of specificity. The operation can be made easier and faster with the aid of the kits, making it accessible to individuals who lack complex operation skills, thus conserving human resources. Furthermore, RPA’s interoperability allows it to be widely utilized in conjunction with other product detection methods [38], such as agarose gel electrophoresis [37], real-time fluorescence probes [39], and lateral flow dipstick (LFD) [40]. When compared to other methods of detection, LFD’s portability, ease of use, visualization, low cost, and quick setup time make it an attractive option [28]. In addition, DNA amplification using RPA and LFD readouts can be completed in under 30 min [41,42]. Recombinase polymerase amplification combined with lateral flow dipstick (RPA-LFD) is currently widely employed in many testing domains, including clinical research [38], food safety research [42], plant disease detection [43], algae detection [44], and others [45]. However, the use of RPA-LFD in detecting HAB species is still in its infancy. As previous studies have not focused on toxic diatom with RPA-LFD approach, we attempted to explore this aspect.

For the management of environmental pollution, it is crucial to explore and develop early detection strategies for HABs that can be conducted on-site using simple methods. Early detection and preventative measures are critical since they can prevent the need for future management efforts. Management decisions based on early detection information may have the possibility of preventing some blooms. However, the primary challenge in early detection lies in the accurate and sensitive detection of algae that are present in low abundance. Effective HAB management and mitigation efforts rely on the availability of time-sensitive and in situ tools for microalgae detection. Thus, this study investigated a specific and sensitive assay for *P. multiseries* using RPA-LFD. The ideal application of RPA-LFD would be the sensitive and rapid detection of *P. multiseries* and the broadening of its application to the field of HAB molecular detection.

## 2. Results

### 2.1. Recombinase Polymerase Amplification Primer and Probe Design, Screening, and System Optimization

Three pairs of RPA primers for the ITS sequence were designed with the primer design software (Primer Premier 5.0) and named PM-RPA-F/R-1, PM-RPA-F/R-2, and PM-RPA-F/R-3 (Table 1). As shown in Figure 1A, when compared with PM-RPA-F/R-1 and PM-RPA-F/R-3, the results of the RPA products from PM-RPA-F/R-2 had a clear and bright band (higher production) and no evidence of non-specific amplification. Therefore, PM-RPA-F/R-2 was selected as the optimal primer set for the subsequent tests in this study.

The amplification temperature and time were optimized to achieve the best amplification effect. Initially, three gradients were set to determine the optimum reaction time for RPA between 10 and 20 min. As shown in Figure 1B, lane 2 was the clearest and brightest. Therefore, 15 min was chosen for RPA. Moreover, to achieve the amplification result as quickly as possible, the RPA amplification temperature was optimized in a temperature range of 35–39 °C. As shown in Figure 1C, lane 1 (35 °C), lane 2 (37 °C), and lane 3 (39 °C) all had bright bands. All three reaction temperatures could be used as conditions for subsequent experiments. Also, in order to reduce the heating-up time, 37 °C was chosen as the final reaction temperature for RPA.

After comparing Figure 2A,B, it was discovered that PM-P-2 had higher specificity and that PM-P-1 had generated a false positive result when the probes had been checked using the LFD strips. Thus, PM-P-2 was selected as the optimal probe for RPA-LFD amplification. Subsequent RPA/RPA-LFD was conducted using the optimal temperature and time that were described above.

### 2.2. Recombinase Polymerase Amplification and Lateral Flow Dipstick Specificity

To ensure the accurate detection of *P. multiseries* in complex environments, the primers and probes were designed to be highly specific in this study. As shown in Figure 3, the genomic DNA of the algal species was used to successfully amplify the ITS sequences using universal primers (TW28 and AB28), alleviating possible interference caused by the degradation of the genomic DNA. The results shown in Figure 3 indicated that the PCR and RPA assays could detect *P. multiseries* algal isolates, but there were no positive results for the other ten algal species. Additionally, all the RPA-LFD assay results agreed with those of the PCR (Figure 4A) and RPA (Figure 4B) assays, which were only visualized using LFD color lines (Figure 4C). Therefore, the results demonstrated that the RPA-LFD assay developed in this study was highly specific for *P. multiseries* and could be used for further experiments.

### 2.3. Recombinase Polymerase Amplification and Lateral Flow Dipstick Sensitivity

The sensitivity levels of the PCR, RPA, and RPA-LFD assays were further tested using ten-fold serial dilutions of the DNA that was extracted from the recombinant plasmids and *P. multiseries* cells, and nuclease-free water was used as a blank control. The detection limit of RPA-LFD was initially determined using the *P. multiseries* genomic DNA (2.0 × 10^4^–2.0 × 10^−3^ pg/μL) in comparison with PCR (Figure 5A) and RPA (Figure 5B). The findings showed that RPA-LFD (Figure 5C) with genomic DNA had a detection limit of 2.0 × 10^2^ pg/μL, which was 10 times greater than that of RPA alone (2.0 × 10^3^ pg/μL) and was the same as that of the PCR (2.0 × 10^2^ pg/μL). Additionally, the sensitivity experiments using the *P. multiseries* recombinant plasmid (1.9 × 10^4^–1.9 × 10^−3^ pg/μL) were performed for the PCR (Figure 6A), RPA (Figure 6B), and RPA-LFD (Figure 6C) assays. The results showed that the detection limit for PCR was 1.9 × 10^0^ pg/μL, while the detection limit for RPA was 1.9 × 10^2^ pg/μL, and the detection limit for RPA-LFD was 1.9 × 10^0^ pg/μL. Overall, the results indicated that the RPA-LFD assay sensitivity for detection of recombinant plasmid DNA was the same as that of the PCR and 100 times higher than the sensitivity of the RPA.

### 2.4. Evaluation of Recombinase Polymerase Amplification and Lateral Flow Dipstick

To evaluate the practicability of the actual detection, the simulated (Figure 7) and field water samples (Figure 8) were analyzed using the RPA-LFD assay developed in this study. As shown in Figure 7, the results showed that the detection limit of the RPA-LFD was 3.52 × 10^1^ cells/mL, which was about 100 times higher than that for RPA and PCR. Moreover, seventeen field water samples and a negative control were tested for the ITS of *P. multiseries*. The results showed that nine field water samples were positive for the ITS as detected by the RPA-LFD assay (Figure 8B). Of the nine positive results, eight originated from the XB seamount and one originated from the XS of the SCS (Figure 8A). Overall, the experimental results indicated that the RPA-LFD assay was suitable for detecting *P. multiseries* in the field.

## 3. Materials and Methods

### 3.1. Algal Species and Cultures

Nine marine algae species were selected for the RPA/RPA-LFD specificity analysis in this study, including *P. multiseries* (NMBguh002-1-1), *Thalassiosira pseudonana* (NMBguh006), *Chaetoceros curvisetus* (NMBguh003-10), *Skeletonema costatum* (NMBguh0042), and *Prymnesium parvum* (NMBjih029), *P. pungens* (NMBguh002-1-2) and *P. delicatissima* (NMBguh002-1-3) were provided by the Microalgae Collection of Ningbo University. Then, *Chaetoceros debilis* (MMDL50116), and *Thalassiosira rotula* (MMDL50319) were supplied by Xiamen University, *Trichodesmium erythraeum* (IMS101) was supplied by the University of Southern California, and *Phaeodactylum tricornutum* (CCMP2561) was supplied by Westlake University. All the marine microalgae were cultured on f/2 medium and placed in a 100 mL flask. The light intensity was 15–20 μmol/(m^2^·s), and the light–dark period was 12:12 h. The temperature was kept at 16 °C. It was a stationary culture, with shaking 1–2 times daily.

### 3.2. DNA Extraction, Polymerase Chain Reaction, Cloning, and Sequencing

The genomic DNAs of all the algal species were prepared using the Ezup Column Bacteria Genomic DNA Purification Kit (Sangon Biotech, Shanghai, China). The DNA concentrations were measured using a Nano-300 microspectrophotometer (Allsheng, Hangzhou, China), and the concentration was 2.0 × 10^4^ pg/μL. The genomic DNA was stored at −20 °C. A pair of PCR primers were designed for the internal transcribed spacer (ITS) of the target sequence of *P. multiseries* and were named Pm-PCR-F (5′-GAGGCTTGGCACTGATACT-3′) and Pm-PCR-R (5′-AGGCATAGAAGTGCTCGTT-3′). The PCR reaction conditions were an initial denaturation at 95 °C for 3 min, 30 cycles of 95 °C for 15 s, 56 °C for 15 s, and 72 °C for 1 min, with a final extension at 72 °C for 5 min. The PCR products were purified and recovered using a High Pure PCR Kit (Magen, Guangzhou, China). The purified PCR products were linked with pMD 18-T vectors (TaKaRa, Dalian, China) and transformed into competent *Escherichia coli* (DH5α). The positive clones were sequenced by Zhejiang Youkang Biotechnology Co. Ltd. (Hangzhou, China).

### 3.3. Recombinase Polymerase Amplification Primers and Lateral Flow Dipstick Probe Design

The ITS sequences that were similar to *P. multiseries* were downloaded from GenBank. The related algal species sequences (mainly the sequences of algal species belonging to the genus *Pseudo-nitzschia*) were compared with the target sequence using the ClustalW program implemented in BioEdit (www.mbio.ncsu.edu/BioEdit/bioedit.html, accessed on 20 May 2023) to identify the specific regions. Then, according to the manufacturer’s instructions, the TwistAmp DNA Amplification Kit (www.twistdx.co.uk, accessed on 21 May 2023) was used to design the RPA primers and LFD probes with Primer Premier 5.0 (www.PremierBiosoft.com, accessed on 25 May 2023). In total, three pairs of primers and two probes were designed. Then, the final optimal primers were selected using RPA, and the optimal probe was selected using RPA-LFD. Finally, the synthesis and labeling of the primers and LFD probe were performed by Zhejiang Youkang Biotechnology Co. Ltd. The primers and probes are shown in Table 1.

### 3.4. Recombinase Polymerase Amplification Conditions Optimization

The TwistAmp Basic Kit (TwistDX, Maidenhead, UK) was used to perform the RPA reactions in a total volume of 50 μL, which contained 29.5 μL of rehydration buffer, 11.2 μL of nuclease-free water, 2.4 μL of forward primer (10 μM), 2.4 μL of reverse primer (10 μM), and 2 μL of genomic DNA that was extracted from *P. multiseries*. Then, vortexing and centrifuging were conducted in a 1.5 mL centrifuge tube. Next, the reaction mixture was transferred into a tube containing lyophilized enzyme pellets that were supplied in the TwistAmp Basic kit, and 2.5 μL MgAc (280 mM) was added to resuspend the pellet. After vortexing and spinning, the solution was incubated in a metal bath (Allsheng, Hangzhou, China) at 39 °C for 20 min. According to the recommended protocol [46], the reaction tubes were flicked and centrifuged after the first 4 min and then re-incubated. Then, the RPA products were purified using a Hipure PCR Pure Kit (Magen, Guangzhou, China) and analyzed through 2% agarose gel electrophoresis. The optimal primers were used to select the temperature and time optimization for the RPA reaction. The temperature of the RPA was set at 35 °C, 37 °C, and 39 °C for 20 min, and a gradient time ranging from 10 to 20 min, increasing by 5 min increments, was used with the optimal temperature. Finally, the optimal selection was conducted according to the principle of saving time and cost.

### 3.5. Recombinase Polymerase Amplification and Lateral Flow Dipstick Assay

The RPA reaction for the RPA-LFD assay was performed in a total volume of 50 µL using a DNA thermostatic rapid amplification kit (Amplification Future, Weifang, China). The reaction mixture consisted of 29.4 μL of rehydration buffer, 11.5 μL of nuclease-free water, 2 μL of forward primer (10 μM), 2 μL of reverse primer tagged with biotin (10 μM), 0.6 μL of LFD probe (10 μM), and 2 μL of genomic DNA that was extracted from *P. multiseries*. The subsequent steps are described above.

To confirm the RPA results, the LFD (Milenia Biotec, Giessen, Germany) was employed to detect the amplicons visually. In brief, 8 μL of RPA reaction product was added directly to 92 μL of assay buffer and mixed thoroughly in a microplate well. Then, the LFD was immersed in the mixture for 3 min. Finally, the result was considered to be positive if both the control line and the detection line were observed. In contrast, the result was considered to be negative if only the control line was observed.

### 3.6. Specificity and Sensitivity Analysis

A total of nine algal species were used as templates for the specificity tests, with ddH_2_O as the template for the negative control. The algal genomic DNA was extracted using the method described in Section 2.2. All the algal DNA templates were used to perform an RPA-LFD assay under the optimal RPA conditions and PCR amplification. Before performing the specificity study, the universal primers TW81 (5′-GGGATCCGTTTCCGTAGGTGAACCTGC-3′) and AB28 (5′-GGGATCCATATGCTTAAGTTCAGCGGGT-3′) [47] were used to perform PCR amplification to validate the integrity of the genomic DNA.

The sensitivity was determined using the genomic DNA from *P. multiseries* and the recombinant plasmid inserted fragment of the ITS sequence of *P. multiseries*, which were extracted using a Pure Plasmid Mini Kit (CWBIO, Suzhou, China). Then, the ddH_2_O was used to dilute the genomic DNA and the recombinant plasmid of *P. multiseries* in a ten-fold serial dilution of eight gradients. The genomic DNA concentration ranged from 2.0 × 10^4^ to 2.0 × 10^−3^ pg/μL, and the recombinant plasmid DNA concentration ranged from 1.9 × 10^4^ to 1.9 × 10^−3^ pg/μL. The ten-fold serially diluted genomic DNA was used to detect the sensitivity of the PCR, RPA, and RPA-LFD, while the plasmid was used as a template and the ddH_2_O was used as a blank control. The protocols for the PCR, RPA, and RPA-LFD were the same as mentioned above. The sensitivity of the RPA-LFD was evaluated by comparing the lower limits of detection of the PCR, RPA, and RPA-LFD.

### 3.7. Recombinase Polymerase Amplification and Lateral Flow Dipstick Test of the Simulated and Natural Samples

The practicability of RPA-LFD in this study was verified by testing the simulated and natural samples. The density of algal cells was counted using a counting chamber with a light microscope (Axioplan; Carl Zeiss, Jena, Germany). Algal cells (3.52 × 10^4^ cells/mL) of *P. multiseries* were obtained by centrifugation at 12,000× *g* for 2 min, and they were diluted at a ten-fold gradient into new water samples. Then, the genomic DNA of the samples was extracted using the same protocol as mentioned above. In addition, the natural seawater samples that were used for this experiment were obtained from the SCS in August–September, 2021. The detailed sampling sites were described by Huang et al. (2023) [2]. In brief, among these sampling sites, XS4 to XS9 were located in the Xisha (XS) sea, and XS8 and XS9 were located near the mouth of the Pearl River Estuary. XB1 to XB20 were located in the seamount regions of Xianbei (XB) in the SCS. At each sampling site, 2 L surface water (−5 m) was sampled using a rosette sampler equipped with a SeaBird CTD system (Ocean Test Equipment, Inc. Fort Lauderale, FL, USA). Water samples were firstly pre-filtered using 200 μm mesh to remove large suspended solids, larger zooplankton, and phytoplankton, followed by a secondary filtration through a 0.2 μm polycarbonate membrane (Millipore, Billerica, MA, USA) using a vacuum filtration pump with negative pressure below 50 kPa. The filter membranes were transferred into tubes and were then snap-frozen in liquid nitrogen and stored at −80 °C until DNA extraction. The same protocol for DNA extraction as mentioned above was used. After the RPA-LFD test of the simulated and natural samples, in contrast with PCR, the practicability of RPA-LFD in this study was verified.

## 4. Discussion

Several studies focusing on the molecular detection of *Pseudo-nitzschia* species have targeted the 28S and 18S ribosomal RNA (rRNA) gene sequences [19,48]. However, cryptic species cannot always be resolved with these genes, and more variable regions, such as the ITS regions, may be more appropriate. The ITS sequence was chosen as the target sequence of qPCR, RAA-LFD, LAMP-LFD, and RPA-LFD, as it is an effective molecular marker that is capable of detecting various HAB species accurately [26,27,49]. The choice of appropriate target sequences is essential for a molecular detection approach. The ITS sequence includes the 3′ end of ITS1, 5.8S, and the 5′ end of ITS2, and, when compared with the large subunit rRNA and small subunit rRNA sequences as molecular markers, it has a higher specificity for designing primers. For example, Huang et al. (2019) [4] and Fu et al. (2019) [44] designed primers that targeted ITS for specific qPCR and RPA-LFD detections of the marine toxic dinoflagellate *Karlodinium veneficum*. Additionally, Andree et al. (2011) [21] created an ITS ribosomal DNA database for the development of a qPCR assay for *Pseudo-nitzschia* spp. Furthermore, the ITS level of conservation has been useful for inter- and intraspecific population studies of *Pseudo-nitzschia* spp. [49]. Furthermore, in this study, the generated primers and probe were submitted to NCBI for Primer BLAST’s online specificity checking. Thus, ITS was chosen as the target sequence to develop a highly specific and practical approach for *P. multiseries* detection using RPA-LFD.

In the procedure of developing an assay, optimizing the detection conditions is a vital stage. To maximize the effectiveness of RPA-LFD detection, it is important to think about the parameters separately from the primers and probes. This includes the temperature and duration of the amplification process to increase the amplification effectiveness, shorten the process, save costs, and ensure that the RPA products produce a visible red band at the test line on the LFD strips [28]. All the aforementioned RPA-LFD systems were tested in this study at a temperature of 37 °C for 15 min, which required minimally complex instruments and fulfilled the stability and sensitivity requirements of in-field applications. The RPA-LFD assay sensitivity for detection of recombinant plasmid DNA was 100 times more sensitive than that of RPA, displaying a detection limit of 1.9 × 10^0^ pg/μL, and the RPA-LFD assay sensitivity for detection of genomic DNA was 10 times more sensitive than that of RPA, displaying a detection limit of 2.0 × 10^2^ pg/μL. To date, some RPA-LFD approaches have been developed for the detection of toxic marine microalgae. For example, Fu et al. (2019) [44] used RPA-LFD to detect *K. veneficum* with a detection limit of 10 ng/μL. Zhang et al. (2022) showed that the detection limit of RPA-LFD for *Chattonella marina* was as low as 9.5 × 10^−1^ ng/μL [49]. In contrast to these studies, our detection limit was even lower. Moreover, the overall time for detection was only 18 min, which was much faster than their times of 35 and 40 min, respectively [44,49,50]. In addition, the method of qPCR developed for the detection of *Pseudo-nitzschia* species by Andree et al. (2011) takes at least 5 h for the entire analysis to be completed [21]. Therefore, although several molecular approaches have previously been used to identify *Pseudo-nitzschia* species and have high specificity and sensitivity [19,20,21], the method that was developed in this study is superior in terms of cost, speed, and convenience. In conclusion, the RPA-LFD that was established in this study provides a novel detection approach for *P. multiseries,* with rapid and intuitive detection outputs and excellent specificity and sensitivity.

Recombinase polymerase amplification is performed at a single reaction temperature, minimizing the need for costly and precise equipment. It also has the advantages of being rapid, easy, and inexpensive, making it ideal for POCT. However, because of the method itself, it still has some defects. Such as RPA reactions produce more non-specific amplifications relative to PCR, a difficult phenomenon to avoid [28,51]. Therefore, it is necessary to screen primers in order to minimize the interference of non-specific amplification for subsequent experiments. RPA is an emerging method, and many researchers have conducted studies regarding the optimization of this method, including specificity enhancement, stirred conditions, and additional methods of detection such as RPA combined with LFD [52,53,54]. In recent years, several scientists have coupled Clustered Regularly Interspaced Short Palindromic Repeats (CRISPR) with RPA to develop an innovative platform for the development of immediate and accurate nucleic acid detection [55,56,57,58]. By combining RPA with a CRISPR-Cas detection system, both SHERLOCK [55] and DETECTR [59] enable nucleic acid detection with attomolar sensitivity for clinical applications. This system also has the potential to be simplified, with Gootenberg et al. (2018) developing an LFD for the visual readout of viral DNA [60]. The combination of CRISPR with RPA resulted in remarkable specificity and sensitivity for the RPA assay detection, which has been successfully used in a variety of fields, such as pathogen detection, genotyping, and disease monitoring [58,60,61]. Given that there are currently few applications for detecting HAB species, there is great potential for development. In the future, it would be very beneficial to develop an approach using RPA-CRISPR-LFD for the rapid on-site detection of HAB species. Overall, the detection system that was identified in this study may help to reduce the current challenges and assist with the early detection and management of blooms due to its robustness and sensitivity, even in the presence of potential inhibitors.

## 5. Conclusions

The *Pseudo-nitzschia multiseries* is a toxic algae that, if it blooms, threatens human health and marine life and leads to major economic losses. In this study, the RPA-LFD assay that was established is a rapid, sensitive, and visual detection system for *P. multiseries.* The whole RPA-LFD approach can be completed in about 18 min (the DNA extraction time is not included), the RPA-LFD assay sensitivity for detection of recombinant plasmid DNA was 100 times more sensitive than that of RPA, and the RPA-LFD assay sensitivity for detection of genomic DNA was 10 times more sensitive than that of RPA. The results demonstrated that it is an effective and practical tool for monitoring HAB species. Thus, the efficiency and usability of RPA-LFD indicate that it is promising for future monitoring. The outcomes of this study encourage the shift from traditional laboratory-based detection to on-site detection to facilitate early warning and monitoring of HABs. This may considerably increase the chances of preventing algal bloom outbreaks in water and the subsequent negative impacts on the environment, people’s livelihoods, and the economy. The proposed RPA-LFD approach will provide a model for on-site early warning detection of HAB in the field.

## Figures and Tables

**Figure 1 ijms-25-01350-f001:**
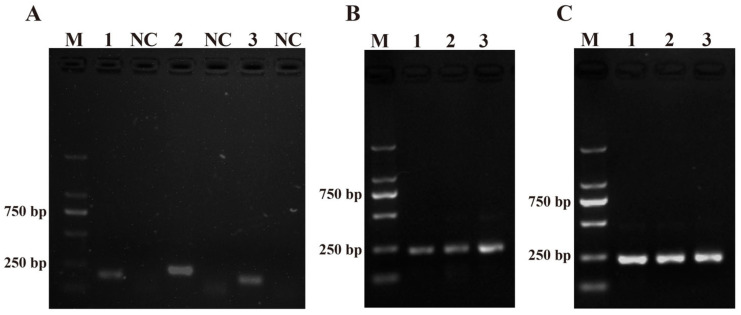
Primers and system optimization. (**A**) The results of the primers for *Pseudo-nitzschia multiseries*. M: DL2000 DNA marker; 1, 2, and 3: positive control of the primers 1–3; NC: negative control of the primers 1–3. (**B**) Optimization of the amplification time. 1: 10 min; 2: 15 min; 3: 20 min. (**C**) Optimization of the amplification temperature. 1: 35 °C; 2: 37 °C; 3: 39 °C.

**Figure 2 ijms-25-01350-f002:**
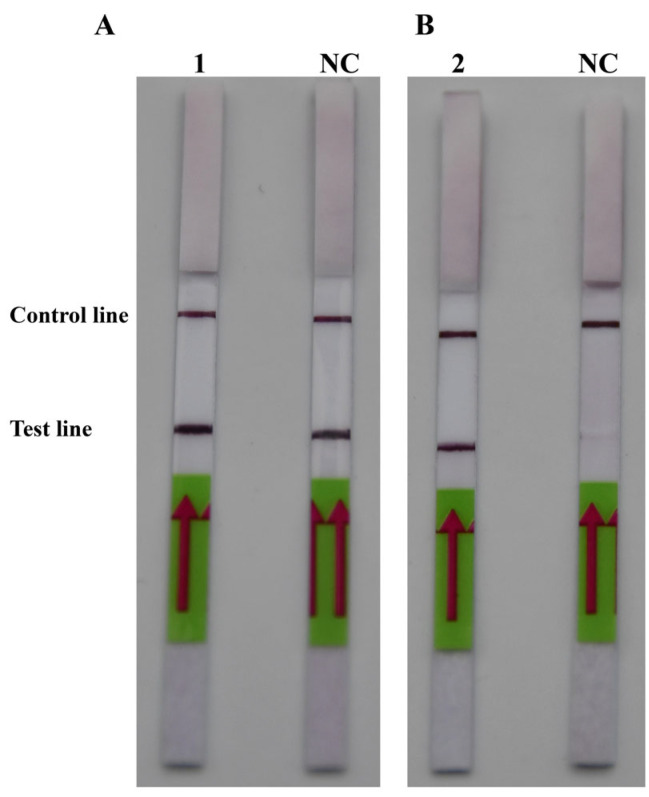
The optimal probes. (**A**) 1: PM-P-1; NC: negative control of probe PM-P-1. (**B**) 2: Probe PM-P-2; NC: negative control of probe PM-P-2.

**Figure 3 ijms-25-01350-f003:**
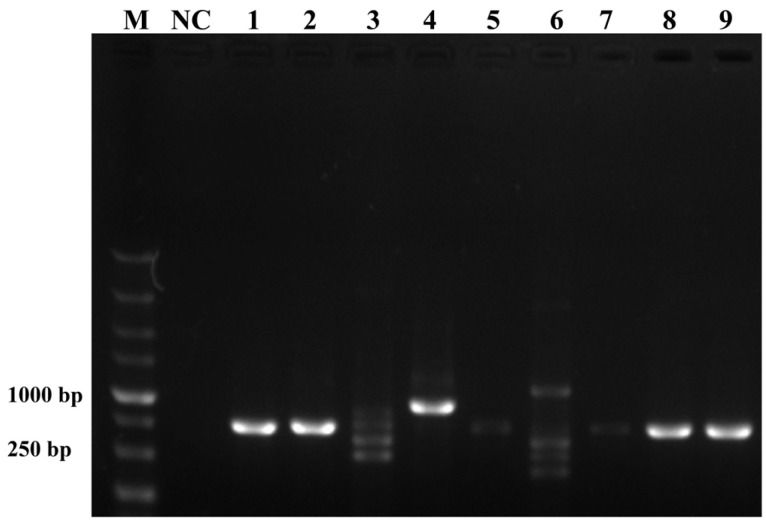
Universal primers for the amplification of the internal transcribed spacer sequences. M: DL5000 DNA marker; NC: negative control; Lane 1: *Prymnesium parvum*; Lane 2: *Thalassiosira pseudonana*; Lane 3: *Chaetoceros debilis*; Lane 4: *Phaeodactylum tricornutum*; Lane 5: *Pseudo-nitzschia multiseries*; Lane 6: *Trichodesmium*; Lane 7: *Skeletonema costatum*; Lane 8: *Thalassiosira rotula*; Lane 9: *Chaetoceros curvisetus.*

**Figure 4 ijms-25-01350-f004:**
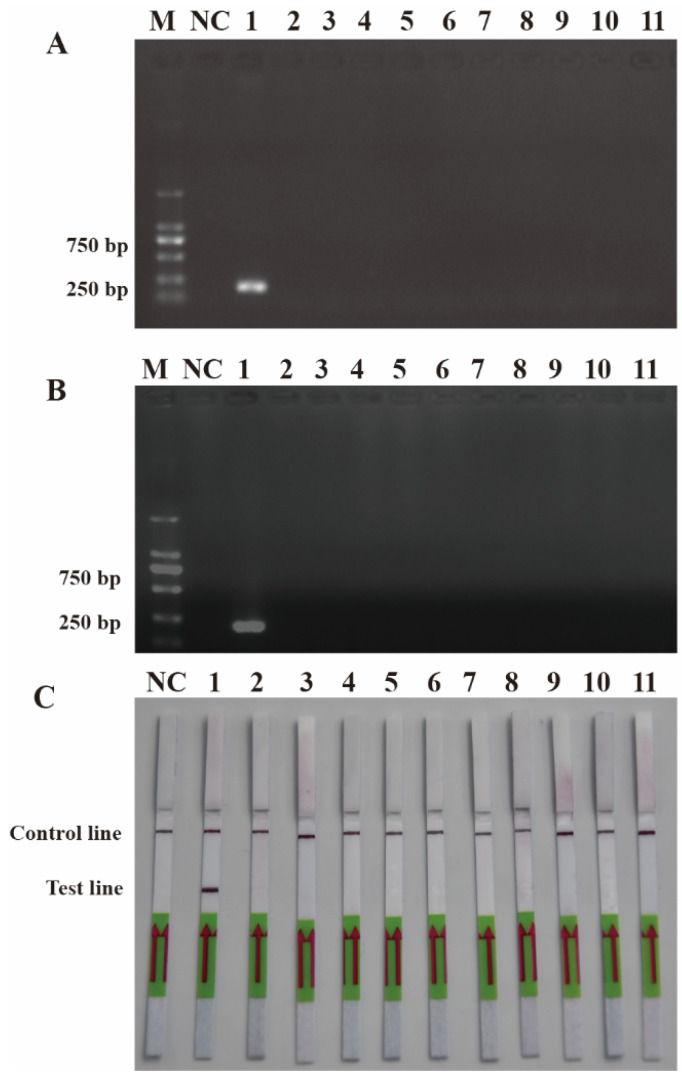
Specificity validation. (**A**) Polymerase chain reaction; (**B**) recombinase polymerase amplification; (**C**) recombinase polymerase amplification with lateral flow dipstick. M: DL2000 DNA marker; NC: negative control; Lane 1: *P. multiseries*; Lane 2: *T. pseudonana*; Lane 3: *C. debilis*; Lane 4: *P. tricornutum*; Lane 5: *P. parvum*; Lane 6: *Trichodesmium* sp.; Lane 7: *S. costatum*; Lane 8: *T. rotula*; Lane 9: *C. curvisetus*; Lane 10: *P. pungens*; Lane 11: *P. delicatissima.*

**Figure 5 ijms-25-01350-f005:**
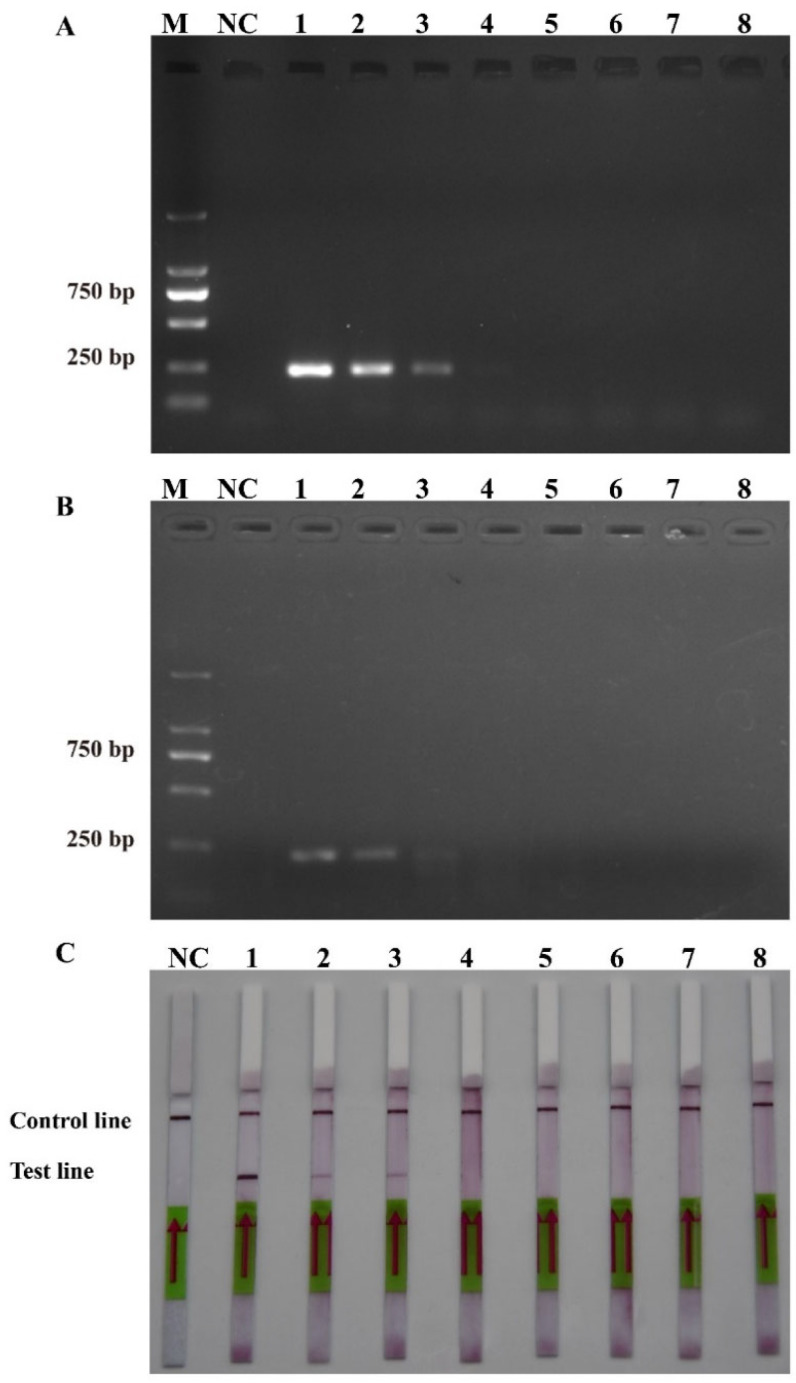
Comparison of the sensitivities of the polymerase chain reaction (**A**), recombinase polymerase amplification (**B**), and recombinase polymerase amplification with lateral flow dipstick (**C**) with the genomic DNA (gDNA). M: DL2000 DNA marker; NC: negative control; Lanes 1–8: gDNA concentration range from 2.0 × 10^4^ to 2.0 × 10^−3^ pg/μL.

**Figure 6 ijms-25-01350-f006:**
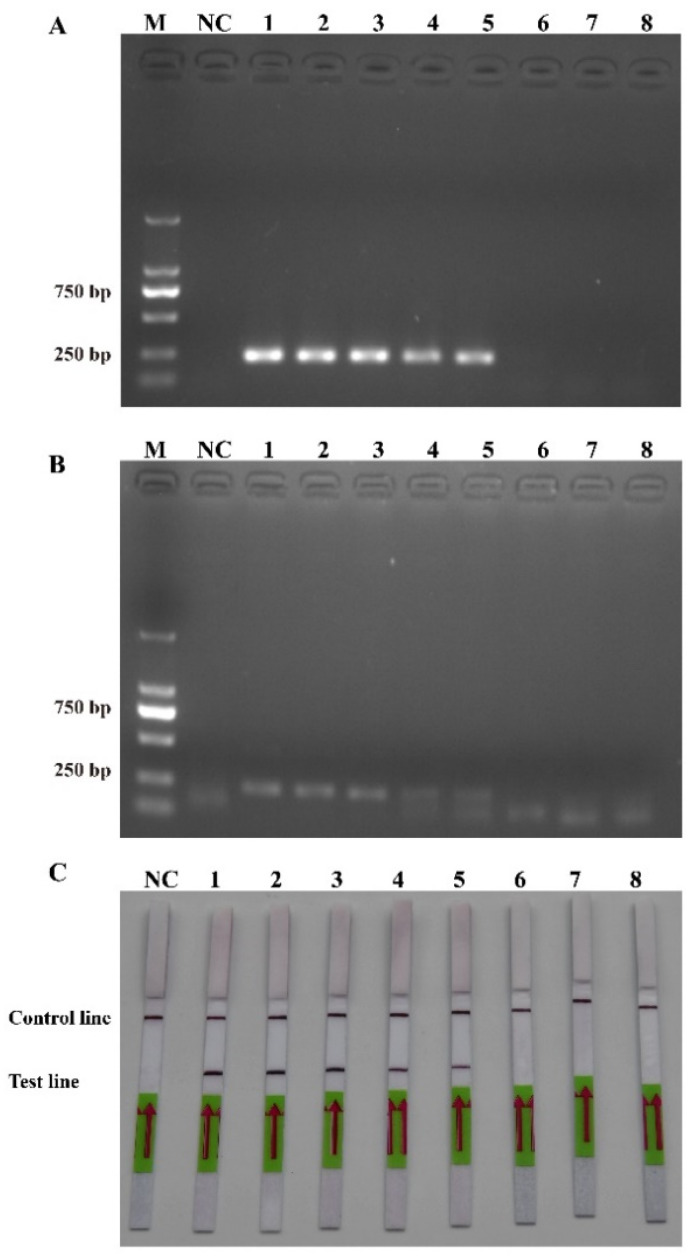
Comparison of sensitivities of PCR (**A**), RPA (**B**) and RPA-LFD (**C**) with recombinant plasmid. M: DL2000 DNA marker; NC: negative control; Lanes 1–8: recombinant plasmid concentration range from 1.9 × 10^4^ pg/μL to 1.9 × 10^-3^ pg/μL.

**Figure 7 ijms-25-01350-f007:**
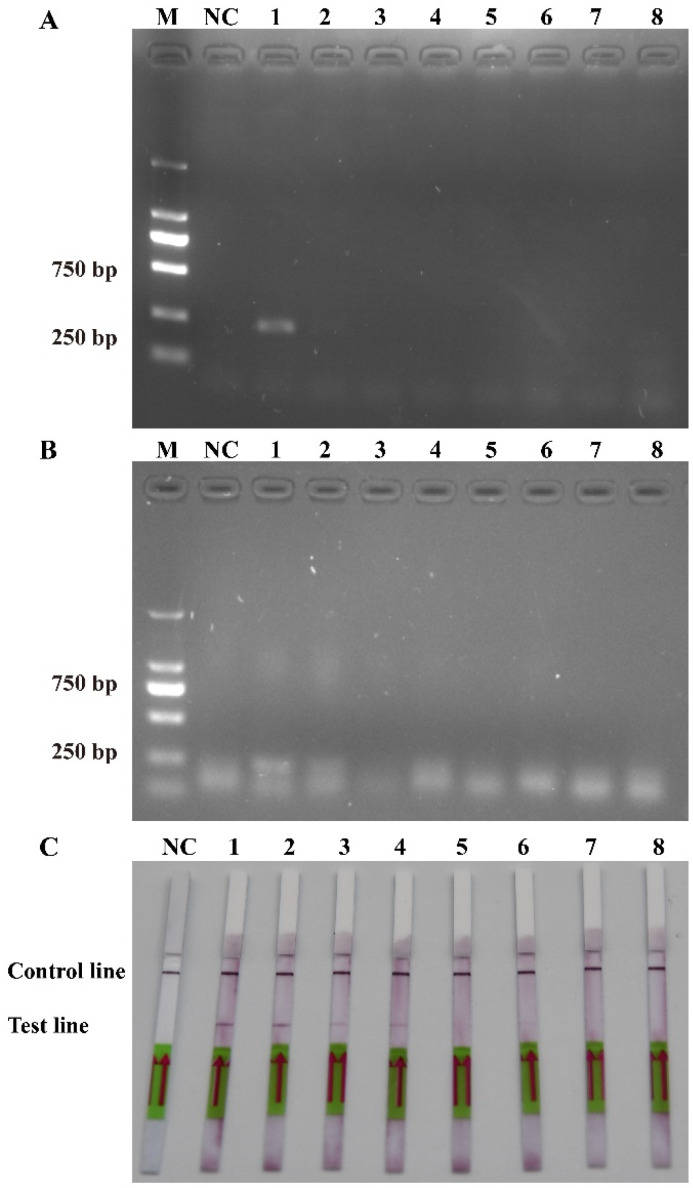
Comparison of the sensitivities of the polymerase chain reaction (**A**), recombinase polymerase amplification (**B**), and recombinase polymerase amplification with lateral flow dipstick (**C**) with the genomic DNA of the simulated sample. M: DL2000 DNA marker; NC: negative control; Lanes 1–8: *Pseudo-nitzschia multiseries* cell concentration ranging from 3.52 × 10^4^ to 3.52 × 10^−3^ cells/mL.

**Figure 8 ijms-25-01350-f008:**
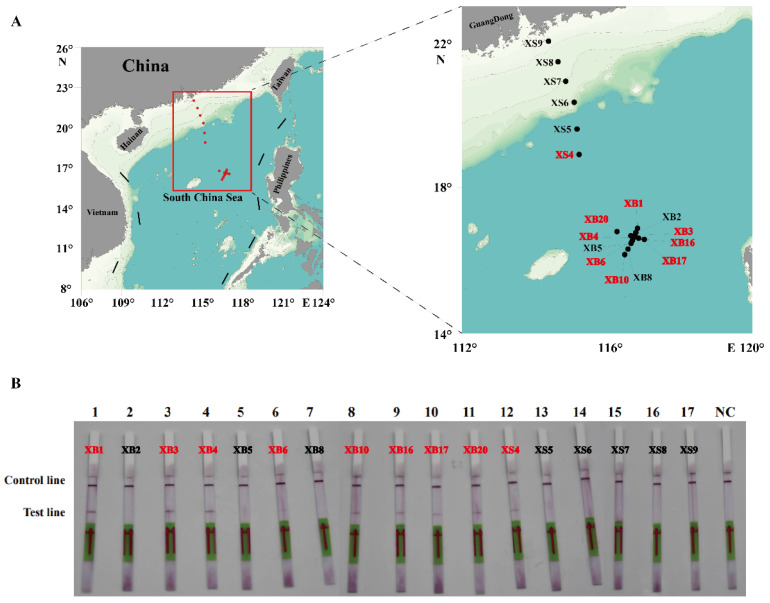
The detection of the recombinase polymerase amplification with lateral flow dipstick assay for the field water samples from the South China Sea. (**A**) Geographical distribution of the water samples in this study. The results of the positive samples are indicated in red font. (**B**) The results are displayed on the test strips. NC: negative control.

**Table 1 ijms-25-01350-t001:** The design of primers and probes for *Pseudo-nitzschia multiseries*.

Primer	Sequence (5′-3′)	GC (%)	Amplification Length (bp)
PM-RPA-F-1	GTTCCCACAACGATGAAGAACGCAGCGAAAT	48.4	187
PM-RPA-R-1	AGTCAAAGCCAAAACAACCAGCAGCCAGCAC	51.6	
PM-RPA-F-2	CCTCGTGCTGGCTGCTGGTTGTTTTGGCTTT	54.8	235
PM-RPA-R-2	AGGCATAGAAGTGCTCGTTCCATCAGTTTCA	45.2	
PM-RPA-F-3	AACGATGAAGAACGCAGCGAAATGCGATACGT	46.9	190
PM-RPA-R-3	GCAATAGTGCCAGTCAAAGCCAAAACAACCAG	46.9	
PM-P-1	[FAM] GTGCATAGACGTGGAAGGCTTGACCTGTCTAGTT [dSpacer]AAGACGGCGTTGACAC[C3-Spacer]		
PM-P-2	[FAM] GCCTGTCTCTGCTTAAGTTCTACTGTATAG [dSpacer]ACGTGCATAGACGTG[C3-Spacer]		

Note: Biotin was tagged to the 5’-end of the PM-RPA-R-2. The 5’-end of the PM-P-1 and PM-P-2 was labeled with a carboxy fluorescein (FAM), the middle of the probe was labeled with a tetrahydrofuran (THF) site, and the 3’-end was labeled with a C3-spacer blocking group.

## Data Availability

All data generated or analyzed during this study are included in this published article.

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
