# Peer review of "Recombinase Polymerase Amplification Combined with Lateral Flow Dipstick Assay for the Rapid and Sensitive Detection of Pseudo-nitzschia multiseries"

_ijms, 2024, doi:10.3390/ijms25021350_

Round 1
Reviewer 1 Report
Comments and Suggestions for Authors
The article by Yao et al addresses an important issue of practical importance. The development of methods for monitoring algae that are toxic to humans and animals is a very important task.
The authors not only optimized the method for detecting toxic Pseudo-nitzscia multiseris and significantly increased its sensitivity, but also tested it on natural samples.
In my opinion, the article is more suitable for Toxics journal, since it will be of greatest interest to the audience of this journal.
It is advisable to indicate the DOI for references in the reference list.
Author Response
Reviewer: #1
The article by Yao et al addresses an important issue of practical importance. The development of methods for monitoring algae that are toxic to humans and animals is a very important task. The authors not only optimized the method for detecting toxic Pseudo-nitzscia multiseris and significantly increased its sensitivity, but also tested it on natural samples. In my opinion, the article is more suitable for Toxics journal, since it will be of greatest interest to the audience of this journal. It is advisable to indicate the DOI for references in the reference list.
Response: We thank the reviewer for the appreciation of work, and the value of our data and analysis. The aim of this study was to develop a recombinase polymerase amplification (RPA) combined with a chromatographic lateral flow dipstick (LFD) assay to rapidly and specifically detect the Pseudo-nitzschia multiseries. The P. multiseries is a diatom species that is found worldwide and is responsible for extensive blooms and death of fish, causing serious negative impacts on the ecological environment. For the prevention and management of environmental pollution, it is crucial to explore and develop early detection strategies for HABs on-site using simple methods. In this study, the P. multiseries is a toxic algae, we did not focus on the study of toxins. We developed an early warning and monitoring approach based on molecular biology method. Furthermore, our research content aligns with the submission scope of the International Journal of Molecular Sciences. Therefore, we think it is more suitable for us to submit the article to IJMS rather than Toxics journal. In addition, we have added the DOI of references in the reference list based on the suggestion of the reviewer. Revised details are marked in red in the revised version of our manuscript.
Reviewer 2 Report
Comments and Suggestions for Authors
The authors developed a highly sensitive method to detect Pseudo-nitzschia sp. based on recombinase polymerase amplification and lateral flow dipstick. The authors optimized the temperature and duration conditions and assessed the suitability of the test with natural samples. Theoretically, the test permits detection of Pseudo-nitzschia at fairly low abundances (35 cel mL-1). The manuscript is well written and the experiments and result description are sound. I have a few doubts regarding to the replication level of the assays and the samplings and analyses of the field samples that I think the authors should clarify in order to improve the manuscript. Some suggestions for modifying the text are also included below.
-Line 16. I think that it should be “is” instead of “are”.
-Line 58. I do not understand why shellfish are “a potential vector for algal toxins”. Please, explain or change the expression.
-Line 210. The method used to cell quantification should be indicated.
-Lines 215-219. Sampling depth, method used to sample collection and procedure used for concentrating cells from these samples (if it was done) should be described. How these samples were managed and afterwards the DNA was extracted is not mentioned.
-Line 234. Were statistical tests used to assess the significance of the differences among treatments? The replication level of these assays must be indicated.
-Lines 302-303. The presence of Pseudo-nitzschia sp in the positive samples must be tested by some other method in order to be sure that they contained cells of this genus. Probably, the best option is comparing the results of the RPA-LFD with microscope counts.
-Lines 350-354. Here, detection limits of the RPA-LFD method based on gDNA are discussed. I wonder if these limits can be also expressed (and discussed) in term of cell abundance.
Author Response
Reviewer(s)' Comments to Author:
Reviewer: #2
The authors developed a highly sensitive method to detect Pseudo-nitzschia sp. based on recombinase polymerase amplification and lateral flow dipstick. The authors optimized the temperature and duration conditions and assessed the suitability of the test with natural samples. Theoretically, the test permits detection of Pseudo-nitzschia at fairly low abundances (35 cell mL-1). The manuscript is well written and the experiments and result description are sound. I have a few doubts regarding to the replication level of the assays and the samplings and analyses of the field samples that I think the authors should clarify in order to improve the manuscript. Some suggestions for modifying the text are also included below.
Response: We thank the reviewer for the enthusiasm towards our results and our manuscript. We also would like to thank the reviewer for the constructive comments regarding our manuscript. Those comments are all valuable for improving our manuscript. Revised details are marked in red in the revised version of our manuscript.
Specific comments:
-Line 16. I think that it should be “is” instead of “are”.
Response: We have replaced the “are” with “is” in the revised manuscript (at line: 22).
-Line 58. I do not understand why shellfish are “a potential vector for algal toxins”. Please, explain or change the expression.
Response: Thanks for the reviewer’s suggestion. The shellfish itself does not produce toxins, but the shellfish may accumulate shellfish toxins in the body by ingesting or living in symbiosis with toxic algae. In addition, we have rewritten the sentence in the revised manuscript (at lines: 65-66) as follows: Shellfish are the dominating cultured organisms and a potential vector for algal toxins.
-Line 210. The method used to cell quantification should be indicated.
Response: Thanks for the reviewer’s suggestion. We have provided the information in the revised manuscript as follows (at lines: 227-228): The density of algal cells was used by a counting chamber with a light microscope (Axioplan; Carl Zeiss, Jena, Germany).
-Lines 215-219. Sampling depth, method used to sample collection and procedure used for concentrating cells from these samples (if it was done) should be described. How these samples were managed and afterwards the DNA was extracted is not mentioned.
Response: Thanks for the reviewer’s suggestion. We have provided the information in the revised manuscript as follows (at lines: 237-244): At each sampling site, 2 L surface water (-5 m), using a rosette sampler equipped with a SeaBird CTD system (Ocean Test Equipment, Inc. Fort Lauderale, Florida, USA). Water samples were firstly pre-filtered using 200 μm mesh to remove large suspended solids, larger zooplankton and phytoplankton, followed by a secondary filtration through a 0.2 μm polycarbonate membranes (Millipore, USA) using a vacuum filtration pump with negative pressure below 50 kPa. The filter membranes were transferred in tubes and were then snap-frozen in liquid nitrogen and stored at -80 °C until DNA extraction. The same protocol for DNA extraction as mentioned above.
-Line 234. Were statistical tests used to assess the significance of the differences among treatments? The replication level of these assays must be indicated.
Response: Thanks for the reviewer’s suggestion. We didn't do a statistical test for three temperature treatments. In this section, the amplification temperature and time were optimized to achieve the best amplification effect. Firstly, we selected the best amplification time. As shown in Fig. 1B, lane 2 was the clearest and brightest. Therefore, 15 min was chosen for RPA. Secondly, we compared the amplification results between different temperatures with the same reaction time. As shown in Fig. 1C, lane 1 (35 ℃), lane 2 (37 °C) and lane 3 (39 ℃) all have bright bands. All three reaction temperatures can be used as conditions for subsequent experiments. Also, in order to reduce the heating-up time, 37℃ was chosen as the final reaction temperature for RPA. We have rewritten these sentences in the revised manuscript as follows (at lines: 262-265): As shown in Fig. 1C, lane 1 (35 ℃), lane 2 (37 °C) and lane 3 (39 ℃) all have bright bands. All three reaction temperatures can be used as conditions for subsequent experiments. Also, in order to reduce the heating-up time, 37 ℃ was chosen as the final reaction temperature for RPA.
Fig. 1 Primers and system optimization. A: The results of the primers for Pseudo-nitzschia multiseries. M: DL2000 DNA marker; 1, 2, and 3: positive control of the primers 1–3; NC: negative control of the primers 1–3. B: Optimization of the amplification time. 1: 10 min; 2: 15 min; 3: 20 min. C: Optimization of the amplification temperature. 1: 35 ℃; 2: 37 ℃; 3: 39 ℃.
-Lines 302-303. The presence of Pseudo-nitzschia sp. in the positive samples must be tested by some other method in order to be sure that they contained cells of this genus. Probably, the best option is comparing the results of the RPA-LFD with microscope counts.
Response: We agree with the suggestion of the reviewer. We have observed the positive water samples under microscope. However, species of Pseudo-nitzschia sp. with small sizes and fragile cells could not be properly observed. We could not decide whether it was Pseudo-nitzschia multiseries or not. Thus, we did not carry out comparative studies of these results.
-Lines 350-354. Here, detection limits of the RPA-LFD method based on gDNA are discussed. I wonder if these limits can be also expressed (and discussed) in term of cell abundance.
Response: Thanks for the reviewer’s suggestion. To date, some RPA-LFD approaches have been developed for the detection of marine toxic microalgae. But the main thing was to develop a rapid, specific and sensitive qualitative assay rather than a quantitative assay of cells. Therefore, the discussions of the detection limits of the RPA-LFD methods are based on gDNA.
Round 2
Reviewer 1 Report
Comments and Suggestions for Authors
To identify potentially toxic microalgae Pseudo-nitzschia, the authors chose highly variable ITS regions located between the 28S and 18S ribosomal RNA genes as a molecular marker. Therefore, the method turned out to be highly sensitive and selective. The use of recombinase polymerase amplification makes it possible to carry out the reaction at a relatively low temperature. The introduction contains enough references on the problem of toxic microalgae blooms, including in the study area. The methods are described in sufficient detail, they include additions relative to the first draft of the manuscript. The caption for Figure 8 is too detailed. Station designations (XB1, etc.) are shown on the map and written on test strips. Therefore, there is no need to include them in the caption to the figure.
Author Response
Reviewer: #1
To identify potentially toxic microalgae Pseudo-nitzschia, the authors chose highly variable ITS regions located between the 28S and 18S ribosomal RNA genes as a molecular marker. Therefore, the method turned out to be highly sensitive and selective. The use of recombinase polymerase amplification makes it possible to carry out the reaction at a relatively low temperature. The introduction contains enough references on the problem of toxic microalgae blooms, including in the study area. The methods are described in sufficient detail, they include additions relative to the first draft of the manuscript. The caption for Figure 8 is too detailed. Station designations (XB1, etc.) are shown on the map and written on test strips. Therefore, there is no need to include them in the caption to the figure.
Response: Thanks for the reviewer’s suggestion. We have deleted the redundant information in the caption of Figure 8.